# Ultrasound Triggers Hypericin Activation Leading to Multifaceted Anticancer Activity

**DOI:** 10.3390/pharmaceutics14051102

**Published:** 2022-05-21

**Authors:** Federica Foglietta, Roberto Canaparo, Simone Cossari, Patrizia Panzanelli, Franco Dosio, Loredana Serpe

**Affiliations:** 1Department of Drug Science and Technology, University of Torino, 10125 Torino, Italy; federica.foglietta@unito.it (F.F.); roberto.canaparo@unito.it (R.C.); simone.cossari@unito.it (S.C.); loredana.serpe@unito.it (L.S.); 2Department of Neuroscience Rita Levi Montalcini, University of Torino, 10125 Torino, Italy; patrizia.panzanelli@unito.it

**Keywords:** hypericin, ultrasound, sonodynamic therapy, sonosensitizer, colon cancer, three-dimensional cell cultures, immunogenic cell death

## Abstract

The use of ultrasound (US) in combination with a responsive chemical agent (sonosensitizer) can selectively trigger the agent’s anticancer activity in a process called sonodynamic therapy (SDT). SDT shares some properties with photodynamic therapy (PDT), which has been clinically approved, but sets itself apart because of its use of US rather than light to achieve better tissue penetration. SDT provides anticancer effects mainly via the sonosensitizer-mediated generation of reactive oxygen species (ROS), although the precise nature of the underpinning mechanism is still under debate. This work investigates the SDT anticancer activity of hypericin (Hyp) in vitro in two- (2D) and three-dimensional (3D) HT-29 colon cancer models, and uses PDT as a yardstick due to its well-known Hyp phototoxicity. The cancer cell uptake and cellular localization of Hyp were investigated first to determine the proper noncytotoxic concentration and incubation time of Hyp for SDT. Furthermore, ROS production, cell proliferation, and cell death were evaluated after Hyp was exposed to US. Since cancer relapse and transporter-mediated multidrug resistance (MDR) are important causes of cancer treatment failure, the US-mediated ability of Hyp to elicit immunogenic cell death (ICD) and overcome MDR was also investigated. SDT showed strong ROS-mediated anticancer activity 48 h after treatment in both the HT-29 models. Specific damage-associated molecular patterns that are consistent with ICD, such as calreticulin (CRT) exposure and high-mobility group box 1 protein (HMGB1) release, were observed after SDT with Hyp. Moreover, the expression of the ABC transporter, P-glycoprotein (P-gp), in HT-29/MDR cells was not able to hinder cancer cell responsiveness to SDT with Hyp. This work reveals, for the first time, the US responsiveness of Hyp with significant anticancer activity being displayed, making it a full-fledged sonosensitizer for the SDT of cancer.

## 1. Introduction

Hypericin (Hyp) is a quinone derivative, structurally related to perylene quinones, that derives from *Hypericum perforatum*, known as St. John’s wort [1,2]. Its broad pharmacological activity, which ranges from antidepressant to anti-inflammatory, antimicrobial, and anticancer effects, is well known [3]. Additionally, its use as an inhibitor of β-amyloid fibril formation has also been studied, making applications in the treatment of Alzheimer’s disease a possibility [4].

The anticancer activity of Hyp relies on its phototoxicity as Hyp is probably the most powerful photosensitizer found in nature. Hyp absorbs in the red wavelength region (around 600 nm) and acts via both the type I (electron or hydrogen atom transfer) and type II (energy transfer to molecular oxygen) reactions. Hyp is characterized by properties that are ideal for a photosensitizer, including its highly efficient production of superoxide anions (O_2_^•−^) and singlet oxygen (^1^O_2_) under light exposure and very-low-to-no toxicity when not exposed to light [5,6]. In recent years, it has largely been studied as an anticancer photodynamic agent [7,8]. Moreover, as the accumulation of Hyp is significantly higher in tumor tissue than in normal tissue, it is used as a fluorescence probe for tumor detection in photodynamic diagnosis [1]. Hyp is therefore an effective natural compound alternative to chemically synthesized photosensitizers. 

Photodynamic therapy (PDT) has been applied as adjuvant tumor therapy for more than 40 years [9,10]. When the photosensitizer is excited by a light of appropriate wavelength, it can directly transfer energy to oxygen to produce reactive oxygen species (ROS) and ^1^O_2_ in tumor cells [10], resulting in severe cell alterations [11]. Light-activated Hyp inhibits various mitogen-activated protein kinases, inducing peroxidation in lipid membranes and mitochondrial damage. Consequently, ROS-mediated PDT leads to tumor killing via apoptotic-, necrotic-, and autophagy-associated cell death [10,12].

In the battle against cancer, sonodynamic therapy (SDT) is under investigation as an interesting approach to overcome some of the limitations of PDT, which include the poor penetration depth of light in biological tissues [13,14,15,16]. Moreover, SDT also has the potential to empower traditional anticancer treatment modalities; since many chemotherapeutic agents have serious dose-dependent side effects, the combination with SDT may have clinical relevance. Indeed, SDT requires low doses of responsive chemical compounds known as sonosensitizers that should be used at noncytotoxic concentrations [17].

The single components of SDT (i.e., the sonosensitizer and US) generate marked cytotoxicity only when combined, thanks to their synergistic effect [14]. This is based on the idea that US can induce physical and chemical changes in biological tissues according to US intensity and frequency [13,18]. Specifically, there are three physical mechanisms on the basis of the biological effects induced by US: (i) heat generation, (ii) acoustic cavitation, and (iii) radiation forces and acoustic streaming [19]. For instance, high-intensity focused US is used for hyperthermia-mediated tumor ablation, while low-intensity US can enhance drug uptake (sonoporation), control release by delivery systems, and activate a sonosensitizer for SDT. However, a deep and complete understanding of the mechanisms behind SDT is still lacking. This can be attributed to the complexity of this approach, with it involving a variety of physical, chemical, and biological processes at the same time.

Acoustic cavitation, a phenomenon in which vapor bubbles oscillate and collapse under the influence of varying pressures in a fluid medium and that can subsequently trigger sonomechanical and sonochemical processes, is one of the most important US-induced phenomena involved in SDT [20]. Noninertial (stable cavitation) and inertial (transient cavitation) have been recognized as two specific types of acoustic cavitation activity. In the first, bubbles persist for several acoustic cycles, oscillating around an equilibrium radius, whereas in the second, bubbles grow more quickly, expanding to two to three times their size, and rapidly and violently collapse once they reach a region with a pressure that is above the vapor pressure, generating several physical effects, including flashes of light, so-called sonoluminescence [21]. Although the mechanism of sonoluminescence is still under debate, it seems to be related to sonosensitizer activation, which may also explain why most sonosensitizers that are employed are also photosensitizers [22]. Sonoluminescent light is absorbed by the sonosensitizer, and the compound is activated, passing from the ground state into an excited state, with the process culminating again in the formation of cytotoxic free radicals, which are responsible for SDT-mediated cytotoxicity. SDT has been extensively investigated in both in vitro and in vivo studies, but is still far from achieving clinical application [14].

Preclinical studies of innovative anticancer treatments are traditionally based on in vitro investigations in two-dimensional (2D) cell cultures. While these cancer models are certainly simple and widely used, they are not able to reproduce the complexity of tumor tissues, making the screening of new anticancer treatments somewhat inefficient. In three-dimensional (3D) cell cultures, cells group together to form spheroidal structures that are more closely related to in vivo tumor masses and mimic various in vivo functions [23,24]. For instance, cancer cells that grow in 3D cell cultures interact with the extracellular matrix, which defines the features of the tissue environment, which, in turn, directly affects cell phenotype [25]. In vitro 3D cancer models are therefore a very valuable tool for cancer research and one that can reduce the gap between in vitro and in vivo cancer models, thus aiding the successful translation of innovative therapeutic approaches. In particular, 3D cancer models provide us with the opportunity to perform more realistic studies of minimally invasive approaches, such as PDT and SDT, as they can take into account the limited sensitizer penetration through the extracellular matrix and the presence of hypoxia, which can limit the treatment efficacy of such oxygen-dependent treatments [26].

As the role of the sonosensitizer is pivotal to SDT outcomes, the ability of Hyp to respond to US stimulation merits investigation. Hyp possesses multifaceted anticancer activity as it is also able to elicit an antitumor immune response upon light activation. This can be mainly ascribed to the induction of immunogenic cell death (ICD) [27], as mediated by damage-associated molecular patterns (DAMPs), which include calreticulin (CRT) and high-mobility group protein 1 (HMGGB1), leading to dendritic cell (DC) maturation and antigen presentation [28,29,30]. Moreover, Hyp appears not to be a substrate of P-gp, meaning that the US activation of Hyp, in a combination treatment with chemotherapeutic drugs, has the potential to counteract P-gp-mediated multidrug resistance (MDR) [31]. Cancer relapse, one of the major causes of cancer treatment failure, can be directly correlated to the insurgence of chemoresistance and consequent treatment failure [32]. The transporter-mediated efflux of drugs by cancer cells is present among the many mechanisms of MDR, with many different transporters being involved, especially the ATP-binding cassette (ABC) transporter superfamily [33]. ABCB1 (MDR1 or P-glycoprotein, P-gp), a member of the ABC transporter superfamily that binds amphipathic and lipid-soluble compounds, plays a pivotal role [34,35]. Although three generations of P-gp inhibitors have been developed to face P-gp-mediated MDR in recent years, they have had low clinical impact [36].

The aim of this work is therefore to evaluate Hyp as a US-responsive agent, namely, a sonosensitizer, by investigating the effect of hypericin-mediated sonodynamic therapy (Hyp-SDT) in 2D and 3D cell culture models of colon cancer using the human colorectal cancer cell line HT-29. Specifically, the Hyp’s ability to generate ROS under US exposure will be investigated as well as its ability to induce ICD and to affect MDR colorectal cancer cells.

## 2. Materials and Methods

### 2.1. Hypericin

4,5,7,4′,5′,7′-Hexahydroxy-2,2′-dimethylnaphtodianthrone (MW 504.44 g/Mol) was purchased as powder (Alfa Aesar, Kandel, Germany) and stored at −20 °C, away from light. A 2 mM Hyp stock solution was prepared in dimethyl sulfoxide (Sigma-Aldrich, Milano, Italy) and stored as aliquots at −20 °C. Working solutions were prepared fresh at a concentration of 50 mM in phosphate-buffered saline (PBS), in accordance with Kleemann et al. [37]. The final Hyp solutions, at concentrations suitable for cell experiments, were then prepared in complete cell culture medium.

### 2.2. Cell Cultures

The human colorectal adenocarcinoma cell line HT-29 was purchased from the American Type Culture Collection (ATCC, Manassas, VA, USA). Two-dimensional HT-29 cell cultures were maintained as monolayers in RPMI-1640 medium (Sigma-Aldrich, Milano, Italy), supplemented with 10% fetal bovine serum (FBS) (Lonza, Verviers, Belgium), 100 UI/mL penicillin, 100 μg/mL streptomycin, and 2 mM L-glutamine (Sigma-Aldrich, Milano, Italy). Cells were cultured in an incubator (Thermo Fisher Scientific, Rodano, Italy) with a humidified atmosphere containing 5% CO_2_ air at 37 °C. Three-dimensional HT-29 cell cultures were developed as spheroids by using a 96-multiwell round bottom (U) plate (BRANDplates^®^ BRAND GMBH + CO KG, Wertheim, Germany) that was previously covered with 1.5% agarose (Sigma-Aldrich, Milano, Italy) to avoid a direct cell attachment to the plate. Briefly, agarose was dissolved in PBS, sterilized in an autoclave at 120 °C for 15 min, and then maintained in a hot water bath to prevent agarose solidification. Agarose (60 µL) was then used to coat each U well. The plates were left to cool down at room temperature for 15 min, then covered with parafilm, and placed in the dark for 24 h. HT-29 cells in their exponential phase were detached using a 0.05% trypsin–0.02% EDTA solution, and 5 × 10^3^ cells were seeded in each well in 200 μL of a culture medium. After 4 days, the spheroids were established, and their growth was monitored over time using a Leica DMI4000B fluorescent microscope (Leica Microsystems, Milano, Italy). The spheroid media were then changed every other day. HT-29 multi-drug-resistant (MDR) cells (HT-29/MDR) were obtained in accordance with Riganti et al. [38]. Briefly, 1 × 10^5^ HT-29 cells were seeded into 75 cm^2^ culture flasks (Techno Plastic Products, TPP, Trasadingen, Switzerland) and cultured in RPMI 1640 medium, supplemented with 10% FBS, 100 UI/mL penicillin, 100 μg/mL streptomycin, 2 mM L-glutamine, and 10 pM doxorubicin (Sigma-Aldrich, Milano, Italy), which is a chemotherapeutic drug that can induce the acquired MDR. After 3-cell passages, the doxorubicin concentration in the culture medium was progressively increased, 2.5 times every 3-cell passages, up to 100 nM. HT-29/MDR cells were periodically tested using a flow cytometric assay with calcein-AM (Sigma-Aldrich, Milano, Italy) to confirm the resistant phenotype, as calcein-AM is a P-gp substrate [32]. Briefly, 1.0 × 10^5^ HT-29 cells were incubated with 2 μL of calcein-AM in 1 mL of PBS at 37 °C for 15 min. Cytofluorimetric evaluation was assessed using a C6 flow cytometer (Accuri C6, Milano, Italy) with excitation at 490 nm and emission at 515 nm. Data were analyzed as the integrated mean of fluorescence intensity (iMFI), which is the product of the frequency of calcein-AM-positive cells and the mean fluorescence intensity of cells. Finally, P-gp expression was evaluated via flow cytometry using a mouse anti-P-gp primary antibody (Ab3366, Abcam, Cambridge, UK) at a dilution of 1:200 in 2.0 × 10^5^ cells for 30 min at room temperature. At the end of incubation, cells were washed with PBS and incubated with a secondary antibody goat anti-mouse conjugated with Alexa Fluor^TM^ 647 nm (Ab150119, Abcam, Cambridge, UK) for 1 h at room temperature. After incubation, cells were washed with PBS, and flow cytometry analysis was performed (λ_ex_ = 650 nm and λ_em_ = 665 nm).

### 2.3. Cytotoxicity Evaluation

In order to identify the optimal concentration of Hyp to act as a sonosensitizer, Hyp cell toxicity was first investigated. Briefly, 2.5 × 10^3^ HT-29 cells were cultured in 100 µL of a culture medium in replicates (*n* = 6) in 96-multiwell plates (Techno Plastic Products, TPP, Trasadingen, Switzerland). The medium was removed after 24 h and replaced with a medium at increasing Hyp concentrations (0.001, 0.01, 0.1, 1 and 10 μM). After 24, 48, and 72 h of incubation, the WST-1 reagent (10 μL/100 μL) was added, and plates were incubated at 37 °C in 5% CO_2_ for 1.5 h. Cellular absorbance was measured at 450 and 620 nm (reference wavelength) using a microplate reader (Asys UV340; Biochrom, Cambridge, UK). The WST-1 cell proliferation assay (Roche Applied, Milan, Italy) relies on the reduction of the WST-1 tetrazolium salt to formazan by mitochondrial dehydrogenases. This metabolic reaction produces a color change that is directly proportional to the mitochondrial dehydrogenase amount and reflects cell number [39]. Cell proliferation is expressed as absorbance (Abs), and cytotoxicity is expressed as a percentage (%), according to the following equation: % cytotoxicity = 100 × (absorbance of untreated cells − absorbance of treated cells). In order to confirm the resistant phenotype in the HT-29/MDR cells, the effects of doxorubicin on cell proliferation was investigated using the WST-1 assay. Briefly, 2.5 × 10^3^ HT-29 and HT-29/MDR cells were seeded in 100 µL of a culture medium in replicates (*n* = 6) in 96-multiwell culture plates. After 24 h, the medium was removed and replaced with a medium that contained doxorubicin at 100 μM. After 48 h, the WST-1 reagent (10 μL/100 μL) was added, and the plates were incubated at 37 °C for 1.5 h. Cellular absorbance was measured as previously reported.

### 2.4. Cell Uptake Evaluation

In order to identify the optimal incubation time for the activity of Hyp as a sonosensitizer, the uptake of Hyp by HT-29 cells was investigated in 2D cell cultures using flow cytometry, thanks to Hyp’s fluorescence properties. Briefly, 8 × 10^4^ HT-29 cells were plated in 6-well culture plates, and 24 h after seeding, cells were incubated for 1, 3, 6, 12, and 24 h with Hyp at 0.1 μM. At the end of each incubation time, cells were detached using trypsin, centrifuged, and resuspended in 100 µL of PBS. Cells were then analyzed using flow cytometry (Accuri C6, Milano, Italy), with excitation at 488 nm, in the spectral range >590 nm [40], considering 1 × 10^4^ events. Cells debris with low forward scatter (FSC) and side scatter (SSC) was excluded from the analysis. Results are expressed as iMFI, which is the product of the frequency of Hyp-positive cells and the mean fluorescence intensity of the cells. Data are expressed as iMFI ratio, which is the ratio between the iMFI values of the treated and untreated cells. Hyp uptake by HT-29 cells was investigated in 3D spheroids using flow cytometry. HT-29 spheroids were incubated with 0.1 and 0.2 μM Hyp for 24 h. After the Hyp-incubation period, four spheroids for each condition were collected in Eppendorf tubes, washed twice with PBS, and then trypsinized for 15 min at 37 °C. Cell pellets were then centrifuged, resuspended accurately in 400 µL of PBS, and analyzed by flow cytometry as previously described.

### 2.5. Confocal Microscopy

The intracellular localization of Hyp was investigated via confocal analysis at the concentration selected for the photodynamic and sonodynamic treatments. Briefly, 4 × 10^5^ HT-29 cells were seeded in 24-well plates with glass coverslips on the bottom of the wells. Twenty-four hours after seeding, cells were incubated with Hyp (0.1 μM) for 3, 6, and 24 h. After each Hyp incubation period, cells were washed with PBS and incubated with MitoTracker^TM^ green FM staining 100 nM (Thermo Fisher Scientific, Rodano, Italy) for 45 min at 37 °C in order to detect the mitochondria. The slides were then washed twice with PBS, and the cells fixed with 4% paraformaldehyde (PAF; Sigma-Aldrich, Milano, Italy) for 15 min at room temperature. The slides were then washed again, and glass coverslips were placed on the slides with Fluoroshield™ mounting medium (Sigma-Aldrich, Milano, Italy) to preserve the cells and prevent the rapid photobleaching of the fluorescent probe. Confocal images were acquired using a laser scanning (λ_ex_ = 405 nm diode laser, λ_em_ = 633 nm) confocal microscope (LSM 900, Zeiss, Milano, Italy) with a 40× oil immersion objective using the multitrack mode. Imagines were then analyzed using ImageJ software (version 2.0).

### 2.6. Photodynamic Treatment

HT-29 cells in the exponential growth phase were preincubated for 24 h, with 0.1 μM Hyp under protection from light. After the Hyp incubation period, cells were washed with PBS, trypsinized, and normalized to 5 × 10^5^ cells in 1 mL PBS in polystyrene tubes (TPP, Trasadingen, Switzerland), under protection from light. Hyp-incubated cells underwent to PDT by being exposed to a light beam (LB) using a light-emitting source based on InGaN light-emitting diodes (Cree Inc., Durham, NC, USA) at a central wavelength of 543–548 nm in a dark box. The energy fluency rate of the LB was fixed at 15 mW/cm^2^ (Jelosil, Le Landeron, Switzerland) for 20 min. After PDT treatment, cells were resuspended in a culture medium, and 5 × 10^3^ HT-29 cells were seeded in 100 μL of a culture medium in replicates (*n* = 6) into 96-multiwell culture plates. Cell proliferation was evaluated 24, 48, and 72 h after PDT using a WST-1 assay. HT-29 spheroids were left to grow for 4 days, as previously described, and then incubated with 0.2 μM Hyp for 24 h, under protection from light. After the Hyp incubation period, spheroids were washed with a culture medium prior to treatment. A total of five spheroids per condition were treated in a 48-multiwell culture plate that was previously coated with 1.5% agarose. Spheroids underwent PDT by being irradiated at 54–3548 nm in a dark box, at 15 mW/cm^2^ for 20 min, as previously described. At the end of the treatment, each spheroid was placed into a 96-multiwell culture plate that had been coated with 1.5% agarose to monitor their growth over time.

### 2.7. Sonodynamic Treatment

HT-29 cells in the exponential growth phase were preincubated for 24 h with 0.1 μM Hyp, under protection from light. After the Hyp incubation period, cells were washed with PBS, trypsinized, and normalized to 5.0 × 10^5^ cells in 2.7 mL PBS in polystyrene tubes, under protection from light. The US field was generated using a piezoelectric plane wave transducer (2.54 cm diameter) that was connected to a power amplifier (Type AR 100A250A; Amplifier Research, Souderton, PA, USA) and a function generator (Type 33250; Agilent, Santa Clara, CA, USA). The polystyrene tube (1.0 cm diameter) was connected to the US transducer, thanks to a particular mechanical adaptor filled with ultrapure water. The US field was generated at 1.5 W/cm^2^ and 1.5 MHz in continuous mode for 5 min. After US exposure, cells were resuspended in the culture medium, and 5 × 10^3^ cells were seeded in 100 μL of growth medium in replicates (*n* = 6) into 96-multiwell culture plates. Cell proliferation was evaluated 24, 48, and 72 h after SDT using a WST-1 assay. HT-29 spheroids were left to grow for 4 days, as previously described, and then incubated with 0.2 μM for 24 h, under protection from light. After the Hyp incubation period, spheroids were washed with a culture medium prior to treatment. A total of five spheroids per condition were treated. Spheroids underwent SDT by being exposed to US in 2.7 mL of PBS into a polystyrene tube. The US field was generated at 1.5 W/cm^2^ and 1.5 MHz, in continuous wave mode for 5 min, as previously described. At the end of the treatment, each spheroid was transferred into a 96-multiwell U plate that was previously coated with agarose 1.5%.

### 2.8. ROS Production

Intracellular ROS production after Hyp-SDT was measured using CellROX^TM^ green (Thermo Fisher Scientific, Rodano, Italy) for oxidative stress detection. Briefly, at the end of Hyp incubation for SDT, HT-29 cells were incubated with 500 nM CellROX^TM^ green for 30 min at 37 °C. Following CellROX^TM^ green incubation, cells were washed with PBS, trypsinized, normalized to 5 × 10^5^ in 2.7 mL of PBS, and subjected to US exposure. ROS production was assessed at 1, 5, 15, 30, and 60 min after US exposure using a C6 flow cytometer (λ_ex_ = 488 nm and λ_em_ = 530 nm) by considering 1 × 10^4^ events and excluding cell debris with low FSC and low SSC from the analyses. ROS production is expressed as iMFI ratio to provide information on the ratiometric variation in fluorescence per time point compared with control cells (untreated cells). This ratio was calculated as the difference in the iMFI of treated and untreated cells over the iMFI of untreated cells [41].

### 2.9. Cell Death Assay

SDT-induced cell death, in the HT-29 2D and 3D cell cultures, was evaluated using a SYTOX™ green dead cell stain for flow cytometry (Thermo Fisher Scientific, Rodano, Italy), according to the manufacturer’s instructions. After the Hyp-SDT of the HT-29 2D cell cultures, 5 × 10^5^ cells were seeded in culture flasks for each treatment condition, and they were trypsinized and collected in PBS after 48 h. HT-29 cells were then stained with SYTOX^TM^ green (1 μL /100 μL) for 15 min at 37 °C, under protection from light. After the Hyp-SDT of the HT-29 spheroids, five spheroids were considered for each condition 48 h after treatment. Briefly, spheroids were centrifuged at 1500 rpm for 2 min, the supernatant was removed, and cell spheroid pellets were then washed with 300 µL of PBS. Finally, spheroid pellets were centrifuged for 2 min and trypsinized for 15 min, and then 1 μL of SYTOX^TM^ green was added to 100 μL of PBS in each flow cytometry tube. HT-29 cells were then incubated for 15 min at 37 °C, under protection from light. At the end of the SYTOX™ green incubation, 400 μL of PBS was added, and samples were analyzed using a C6 flow cytometer at 488 nm excitation and 530 nm emission. A total of 1 × 10^4^ events were analyzed to discriminate viable (SYTOX^TM^ green negative) from dead (SYTOX^TM^ green positive) cells. Cell debris with low FSC and SSC was excluded from the analyses. To confirm the SDT-induced cell death in HT-29 spheroids, fluorescence microscopy was also carried out using propidium iodide (PI; Thermo Fisher Scientific, Rodano, Italy), according to the manufacturer’s instructions. Briefly, 48 h after Hyp-SDT, spheroids were washed twice with PBS and then incubated with PI for 20 min at room temperature, under protection from light. At the end of PI incubation, each spheroid was washed with PBS, and images were acquired (λ_ex_ = 540 nm and λ_em_ = 590 nm) using a Leica DMI4000B fluorescent microscope (Leica Microsystems, Milano, Italy). Images were then analyzed using ImageJ software to quantify the PI fluorescence of HT-29 spheroids, and the results are expressed as mean of PI intensity/μm^2^ ± standard deviation.

### 2.10. Calreticulin and High-Mobility Group Box 1 Evaluation

The exposure of calreticulin (CRT) on the HT-29 cell surface was determined 6 h after treatment with Hyp-SDT. Briefly, 5 × 10^5^ cells for each experimental condition were considered and incubated with Alexa Fluor^®^ 488 anti-CRT antibody (ab196158; Abcam, Cambridge, UK), 10 μg/mL, for 40 min at room temperature, under cover from light. At the end of the incubation, cells were washed twice with PBS and then immediately analyzed using a C6 flow cytometer (λ_ex_ = 488 nm, λ_em_ = 530 nm). In the analysis, 1 × 10^4^ events were considered at a medium flow rate, and cellular debris with low FSC and low SSC were discarded. HMGB1 occurrence was determined 24 h after treatment with Hyp-SDT. Briefly, 5 × 10^5^ cells for each experimental condition were considered and incubated with anti-HMGB1 antibody (ab77302; Abcam, Cambridge, UK), 10 μg/mL, for 30 min at room temperature, protected from light. At the end of the incubation, cells were washed once with PBS and then incubated with rabbit F(ab’)2 goat anti-mouse IgG H&L (Alexa Fluor^®^ 647) (ab150119; Abcam, Cambridge, UK), 1:2000, for 1 h at room temperature, under protection from light. At the end of incubation, cells were washed twice with PBS and then analyzed using a C6 flow cytometer (λ_ex_ = 640 nm and λ_em_ = 665 nm). In the analysis, 1 × 10^4^ events were considered at a medium flow rate, and cellular debris with low FSC and low SSC were discarded.

### 2.11. Statistical Analyses

Data are reported as mean values ± standard deviation of three independent experiments. Statistical analysis was carried out by using Prism 9.0 software (GraphPad, La Jolla, CA, USA). According to the experimental design, multiple *t*-test, two-way ANOVA, one-way ANOVA analysis of variance, and Bonferroni’s test were used to calculate the threshold of significance. The statistical significance was set at *p* ≤ 0.05.

## 3. Results

### 3.1. Hypericin Cytotoxicity

To select the Hyp noncytotoxic concentration for sonodynamic experiments, HT-29 cells in 2D were incubated at increasing concentrations of Hyp (0.001, 0.01, 0.1, 1, and 10 μM) for 24, 48, and 72 h. A significant Hyp cytotoxicity was observed starting from 1 µM after 24 (*p* ≤ 0.01) and 48 (*p* ≤ 0.05) h, and the highest Hyp concentration tested (10 µM) resulted in a significant cytotoxicity (*p* ≤ 0.001) over time (Figure 1). Therefore, the maximum Hyp noncytotoxic concentration, 0.1 μM, was selected as the proper concentration for investigating the possible synergistic activity of Hyp in combination with US.

### 3.2. Hypericin Cellular Uptake and Intracellular Localization

To select the best suitable incubation time of Hyp for sonodynamic experiments, the intracellular uptake of 0.1 μM Hyp was evaluated in HT-29 2D cell cultures at different incubation times (1, 3, 6, 12, and 24 h). An increase in Hyp uptake was observed over time up to a maximum level ranging from 6 to 24 h (Figure 2). Since no statistically significant differences were observed among 6, 12, and 24 h, 24 h of Hyp incubation was selected as the most appropriate time for investigating the possible synergistic activity of Hyp in combination with US.

A confocal fluorescence analysis of Hyp in HT-29 cells was then performed to confirm the cytofluorimetric data by establishing Hyp cellular localization, as cellular localization can significantly influence the sonodynamic activity of Hyp. In the same procedure as the cytofluorimetric study of Hyp uptake, HT-29 cells were incubated for 3, 6, and 24 h with 0.1 μM Hyp. By staining mitochondria with an appropriate probe, MitoTracker^TM^ green, overlay imaging analysis revealed that Hyp displayed cytoplasmic distribution, specifically at the mitochondrial level (Figure 3). The mitochondrial localization of Hyp increased over time, in accordance with the cytofluorimetric study of Hyp cellular uptake (Figure 2). Moreover, the accumulation of Hyp clusters close to the cell membrane was observed. 

### 3.3. Hypericin-Induced ROS Generation under US Exposure

Since the ability of Hyp to provoke ROS production under PDT treatment has been reported in the literature [42], we thought it would be interesting to analyze ROS production under US exposure. As observed in Figure 4, slight, but statistically significant, ROS generation was observed in HT-29 cells under the combined treatment of Hyp and US over time (*p* ≤ 0.05), with the maximum being found 5 min after treatment. Moreover, the signal, expressed as iMFI ratio versus Ctrl (i.e., untreated cells), remained almost constant up to 1 h after treatment.

### 3.4. Effects of Photodynamic and Sonodynamic Treatment with Hypericin on HT-29 Cells

As Hyp is a well-known photosensitizer, the effects of PDT with Hyp on HT-29 cell proliferation were evaluated, using the selected noncytotoxic concentration of Hyp (0.1 μM), 24, 48, and 72 h after treatment. A significant reduction in HT-29 cell proliferation was observed only when the HT-29 cells underwent PDT, starting from 48 h after treatment, whereas no significant effect was observed in untreated cells, cells treated with Hyp only, and cells exposed to LB only (Figure 5a). A marked increase in the percentage of dead cells (58.34 ± 6.47%, *p* ≤ 0.001) and a statistically significant decrease in the percentage of live cells (41.66 ± 4.93%, *p* ≤ 0.001), both compared with untreated cells, were observed 48 h after the photodynamic treatment of HT-29 cells with Hyp and LB (Figure 5b).

To verify the responsiveness of Hyp, acting as a sonosensitizer, to US, the effects of Hyp-SDT were evaluated on HT-29 cell proliferation using a Hyp noncytotoxic concentration of 0.1 μM. A statically significant decrease in cell proliferation was observed when HT-29 cells underwent Hyp-SDT, starting from 48 h after treatment (Figure 6a). A significant increase in the percentage of dead cells (33.20 ± 8.97%, *p* ≤ 0.01) and a statistically significant decrease in the percentage of live cells (66.60 ± 4.25%, *p* ≤ 0.01), both compared with untreated cells, were observed 48 h after the sonodynamic treatment of HT-29 cells with Hyp and US (Figure 6b).

### 3.5. CRT and HMGB1 Induced by Sonodynamic Treatment with Hypericin on HT-29 Cells

CRT is an early signal exposed on cancer cell membranes and an important ICD-related DAMP [43,44]. CRT exposure on HT-29 cell surfaces was therefore evaluated 6 h after Hyp-SDT treatment, and a significant increase in the CRT fluorescent signal was observed in the SDT condition compared with untreated cells (Ctrl), which are indicated by a dashed line (Figure 7, *p* ≤ 0.05). Another important ICD hallmark is the release of HMGB1 from the nucleus of dying cells, and this is considered a late signal in the ICD pathway [40]. Therefore, HMGB1 release was evaluated in HT-29 cells 24 h after Hyp-SDT treatment. A significant increase in HMGB1 was observed when HT-29 cells underwent Hyp preincubation, followed by US exposure (Figure 7, *p* ≤ 0.05).

### 3.6. Effects of Sonodynamic Treatment with Hypericin on HT-29/MDR Cells

The acquired MDR in HT-29/MDR cells was verified by cytofluorimetric analysis of calcein-AM, a substrate of P-gp that can highlight differences in its own cellular internalization according to P-gp expression. The results were compared with the ones obtained from the parental cell line, HT-29. HT-29/MDR cells presented a significantly lower internalization of calcein-AM than HT-29 cells (Figure 8a, *p* ≤ 0.01). To confirm the resistant phenotype in HT-29/MDR cells, P-gp expression was also evaluated using a specific P-gp antibody to allow its cytofluorimetric detection. The cytofluorimetric analysis displayed a 2.4-fold increase in P-gp expression of HT-29/MDR cells compared with HT-29 cells (Figure 8b). Moreover, the cytotoxic effect of a chemotherapeutic drug that is a well-known P-gp substrate, doxorubicin [45], was evaluated at a concentration (100 μM) above its IC_50_ value [46]. After 48 h of doxorubicin incubation, a significant decrease in HT-29 cell proliferation was observed, while no significant effect on cell proliferation was observed in HT-29/MDR (Figure 8c).

Moreover, when HT-29/MDR cells underwent sonodynamic treatment with Hyp, a statistically significant reduction in cell proliferation was observed after 48 h (Figure 9, *p* ≤ 0.01) unlike what was observed when incubating the cells with doxorubicin (Figure 8c).

### 3.7. Effects of Photodynamic and Sonodynamic Treatment with Hypericin on HT-29 Spheroids

Prior to performing photodynamic and sonodynamic treatment on HT-29 spheroids, an uptake study of Hyp in HT-29 cells organized into 3D structures was carried out. Since Hyp uptake may be influenced by the multiple layers of cells that characterize the 3D architecture, HT-29 spheroids were incubated with a Hyp concentration (0.2 µM) twice that used in the 2D cell cultures (0.1 µM), for the same incubation time (24 h). As shown in Figure 10, significant Hyp cellular uptake was observed 24 h after the HT-29 spheroids were incubated with 0.2 µM Hyp (*p* ≤ 0.01). It is worth noting that the iMFI ratio induced in HT-29 spheroids by 0.2 µM Hyp was similar to that obtained in the HT-29 monolayers incubated with 0.1 µM Hyp (Figure 2). Therefore, 0.2 µM was chosen as the proper concentration with which to investigate the possible synergistic activity of Hyp and US in HT-29 spheroids.

The effect of photodynamic treatment with Hyp on HT-29 spheroids was first evaluated by analyzing their shape using optical microscopy. HT-29 spheroids that underwent PDT with Hyp showed significant shape modification, with a loss of central density, 48 h after treatment. No modifications in the HT-29 spheroid shape were observed in spheroids treated with Hyp only or those exposed to LB only (Figure 11a). The PDT effect was then evaluated by incubating spheroids with PI to detect dead cells via fluorescence microscopy, 48 h after treatment. Only HT-29 spheroids that underwent PDT showed a significant increase in PI fluorescence compared with untreated spheroids (Figure 11b).

The effect of Hyp-PDT on the HT-29 spheroids was then evaluated by analyzing cell death using flow cytometry. As shown in Figure 12, a statistically significant increase in the percentage of dead cells (33.22 ± 9.81%, *p* ≤ 0.05) and a statistically significant decrease in the percentage of live cells (66.78 ± 12.68%, *p* ≤ 0.05), both compared with untreated spheroids, were observed 48 h after the photodynamic treatment of HT-29 spheroids with Hyp and LB, which is in line with the results of the microscopy analyses (Figure 11).

The effect of Hyp-SDT on the HT-29 spheroids was first evaluated by analyzing their shapes with optical microscopy. HT-29 spheroids that underwent Hyp preincubation, followed by US exposure, showed clear edges, but they were smaller than those of the untreated spheroids, 48 h after treatment (Figure 13a). The SDT effect was also evaluated by incubating spheroids with PI to detect dead cells, 48 h after treatment. A significant increase in PI fluorescence was observed in SDT-treated spheroids compared with untreated spheroids, while they also maintained their external spheroid shape (Figure 13b).

The evaluation of sonodynamic treatment with hypericin on HT-29 spheroids was also carried out by analyzing cell death by flow cytometry. As shown in Figure 14, a marked increase in the percentage of dead cells (44.00 ± 3.50%, *p* ≤ 0.01) and a significant decrease in the percentage of live cells (56.00 ± 4.00%, *p* ≤ 0.01), both compared with untreated spheroids, were observed 48 h after the sonodynamic treatment of HT-29 spheroids with Hyp and US, which is in line with the results of the microscopy analyses (Figure 13).

## 4. Discussion

Hyp has been widely studied as a photodynamic agent in cancer treatment due to its high triplet quantum yield, high cancer selectivity, and low cytotoxicity [47,48]. Moreover, several studies have demonstrated the efficacy of light-activated Hyp in PDT-induced cancer killing [47,48,49]. SDT is evolving as an alternative noninvasive strategy for cancer treatment that relies on the cytotoxicity induced by the combination of sonosensitizer and US [14]. Like PDT, several studies have identified sensitizer-mediated ROS generation as the main mechanism of SDT-induced cancer cell killing as the release of ROS causes oxidative damage in proteins, lipids, and DNA [50,51,52]. Unlike PDT, SDT can reach deep-seated tumor areas, thanks to the high penetration ability of US. While the exact SDT mechanism of action is still under debate, acoustic-cavitation-mediated sonoluminescence is believed to be a key mechanism in generating sonosensitizer-mediated ROS [52,53]. Moreover, the characteristics of the sensitizer is one of the key factors in determining the efficacy of both PDT and SDT, and the physicochemical properties of the sensitizer influence not only the therapeutic outcomes of PDT and SDT but also treatment safety. Furthermore, we must remember that most sonosensitizers derive from photosensitizers that have been widely used in PDT [54]. However, Hyp-mediated SDT has only been studied in the induction of THP-1 macrophage apoptosis for atherosclerosis treatment, leading to autophagy through the AMPK/mTOR pathway [55]. Hyp-SDT reduces lipids in macrophages through the upregulation of the ROS-dependent nuclear translocation of transcription factor EB, which is involved in the autophagy–lysosome pathway [56]. In macrophages, this pathway lowers the amount of intracellular lipids, suppressing foam cell generation and promoting lipid depletion in plaques, thus lowering atherosclerosis progression [55].

In the present work, Hyp was investigated as a sonosensitizer for anticancer SDT in a colon cancer model. The proper Hyp concentration and incubation time for Hyp-mediated SDT were chosen in accordance with data from cytotoxicity and cellular uptake studies. No cytotoxicity was observed in HT-29 cells at Hyp concentrations lower than 1 μM, whereas the highest concentration, 10 μM, exerted significant cytotoxicity over time. Ozen et al. have already observed the effect of Hyp on cell proliferation in human leukemic HL-60 cells, resulting in a time- and dose-dependent cytotoxicity with a half-maximal cytotoxic concentration (IC_50_) of 0.5 μM [57]. In human colorectal cancer HT-29 cells, a half-maximal cytotoxic concentration was not observed up to 100 μM concentration. A noncytotoxic concentration of 0.1 μM was therefore selected as the proper sonosensitizer concentration for the study of the Hyp cytotoxicity triggered in 2D HT-29 cell cultures by US.

Since, in both PDT and SDT, sensitizer-induced damage occurs in proximity to the oxidizing species produced by the excited molecules [48], sensitizer cellular uptake and its intracellular localization are crucial if photodynamic and sonodynamic protocols are to be properly tuned. In HT-29 cells, the maximum level of Hyp intracellular uptake was observed between 6 and 24 h with somewhat of a plateau being reached after 6 h of incubation. Furthermore, upon comparing 2D and 3D cell cultures, similar results were obtained when double the concentration of Hyp (0.2 mM) was used on the 3D cultures to account for the 3D architecture [58]. Although the cellular mechanisms of Hyp uptake remain unclear, the results obtained by Thomas et al. suggest that Hyp might be transported into cells via partitioning, pinocytosis, or endocytosis [59]. Concerning its subcellular redistribution, Hyp localizes in a time-dependent manner in cytoplasmic membranes, particularly accumulating into the membranes of the endoplasmic reticulum, Golgi apparatus, and mitochondria [48,60].

In order to investigate Hyp distribution inside HT-29 cells, confocal microscopy analyses have been performed, confirming the cytoplasmic distribution of Hyp in cells, which increases over time, specifically at the mitochondrial level with peculiar accumulation being observed close to the cell membrane. This last fact may be attributed to Hyp localization in the endoplasmic reticulum. This Hyp localization close to the cell membrane may perhaps explain the observed efficient Hyp cytotoxicity under US stimulation, considering the mechanical nature of US waves and their fundamental interactions with these membranes [61].

Since it has been shown that ROS in cancer cells are strictly connected to apoptosis activation and autophagy in response to SDT [62], a ROS investigation under Hyp stimulation by US has been performed in HT-29 cells. Our data demonstrate a significant increase in ROS generation up to 5 min after treatment, followed by a slow reduction. While we find these data particularly interesting, they are only supported in the literature by the work of Li and colleagues, who observed ROS production in the early first 30 min, with a further extension to 1 h when THP-1 macrophages were treated with Hyp and then exposed to US [56].

Starting with these data, experiments have been performed on both 2D and 3D HT-29 cell cultures to investigate the synergistic action of Hyp and US, with Hyp activity under light exposure also being reported as a reference. Since strong evidence of SDT anticancer efficacy in 2D cancer cell culture models has been reported, experiments on 3D cancer cell models are needed to further substantiate the ability of US to trigger sonosensitizer cytotoxicity. Three-dimensional cell cultures better imitate living tissues, and their use has the potential to lower drug development costs, allowing more efficient drug screening and thereby minimizing failure rates and decreasing animal use during drug discovery [63]. Spheroids are the main 3D cell culture model used in cancer research because they are able to reproduce a great number of the structural, physiological, and biological features of solid tumors [64].

Upon comparing the effects of Hyp-mediated photodynamic and sonodynamic treatments in 2D and 3D HT-29 cell cultures, we observed that US was able to efficiently trigger Hyp cytotoxicity, just like light, leading to a significant decrease in cell proliferation and a marked increase in cell death. As expected, Hyp-PDT in 2D HT-29 cell cultures resulted in significant cytotoxicity starting from 48 h after treatment, along with a significant increase in the percentage of dead cells. Similar results were obtained from HT-29 spheroids, where flow cytometry analyses revealed a significant increase in the percentage of dead cells along with a significant decrease in the percentage of live cells 48 h after Hyp-PDT. The data were also supported by optical and fluorescence microscopy, which showed, after Hyp-PDT, spheroid disaggregation and increased PI staining, respectively.

Hyp-SDT in 2D HT-29 cell cultures also exerted significant cytotoxicity starting from 48 h after treatment and a significant increase in the percentage of dead cells, which were also confirmed by cytofluorimetric data. Interestingly, spheroids treated with Hyp-SDT were found to be smaller than untreated spheroids and showed an increase in PI staining.

Three-dimensional cancer models have recently been noted to efficiently mimic in vivo cancer features, such as growth [65], genomic heterogeneity [66,67], cancer cell migration [68,69,70], or drug resistance [71]. Indeed, cells cultured in 3D can exhibit different responses to drugs compared with cells cultured in 2D [64,72]. The differences in the physiological features of 2D and 3D cultures usually mean that 2D cells are more responsive to drug effects than 3D cells. Indeed, cells in 2D cultures are unable to preserve normal morphology in the way that cells in 3D cultures do. Moreover, the higher responsiveness to drugs of 2D cell cultures may be influenced by differences in the organization of surface receptors compared with 3D cell cultures [59]. In addition, most of the cells in 2D cultures are at the same cell stage, whereas cells in 3D cultures are usually at a variety of cell stages, as are cells in vivo [63]. Therefore, 3D systems can promote the development of new drug candidates and novel therapeutic effects. For instance, it has been reported that 3D cultures are more suitable models than 2D cultures to investigate the efficacy of treatment, such as PDT [73,74]. However, there are few papers that investigate SDT in 3D cell cultures [75,76] that could represent a suitable tool to optimize this US-based treatment before approaching in vivo models.

In this work, experiments conducted in 2D and 3D cell cultures gave similar results, in terms of treatment response, in experiments on Hyp activation by light and US exposure. Therefore, once the proper sensitizer concentration was chosen according to the type of cell culture, the use of physical agents (e.g., light and US) did not seem to be influenced as much by interactions with 2D or 3D structures as by the intracellular localization of the sonosensitizer.

Recent investigations on SDT have highlighted its ability to induce ICD [77] similarly to PDT, which is already recognized as a treatment modality that can induce this peculiar type of cell death [29,78]. SDT has been shown to work in a comparable way to PDT, relying on ROS production as well, whereas US has also shown immunomodulatory action [79]. However, the exact mechanisms underlying ICD induction are still poorly understood. One formulated hypothesis is that oxidative stress may be the pivotal mechanism that makes cancer cells more immunogenic compared with cancer cells killed by other mechanisms. In accordance with this idea, it has been reported that some DAMPs that are associated with oxidative stress, such as peroxidized phospholipids, may be recognized as immunogenic by the immune system [80]. Therefore, the activation of antitumor immunity contributes to killing primary tumor cells as tumor cells even at isolated locations [81]. Given the pivotal role of immune responses, exploiting anticancer immune responses may pave the way to new possibilities for long-lasting tumor control.

Some studies have demonstrated that Hyp directly accumulates in the endoplasmic reticulum and mitochondria [6], and it is able, under proper light exposure as a nonporphyrin photosensitizer in PDT, to generate high levels of ROS along with a strong anticancer immune response [82,83]. Therefore, to assess whether our SDT approach may have immunological features, an ICD investigation in our experimental setup was evaluated, and CRT and HMGB1 occurrence was observed after Hyp was exposed to SDT. The statistically significant presence of both ICD markers, CRT and HMGB1, was detected at 6 and 24 h, respectively. Moving to the literature, several in vitro and in vivo works have already reported the role of SDT in inducing ICD, although the sensitizer used belonged to the porphyrin family in most cases [84]. In our work, we have demonstrated, for the first time, that the sonodynamic stimulation of Hyp induces ICD marker release, suggesting the existence of a strict correlation between SDT-Hyp and ICD induction.

Other than stimulating an anticancer immune response, hindering the effect of acquired MDR is another important strategy when facing cancer treatment failure. SDT has the potential to also be effective on cells displaying transporter-mediated multidrug resistance (MDR) by exploiting the cytotoxicity of sonosensitizers that are not substrates of MDR transporters [85]. Moreover, it has been reported that SDT can decrease the expression levels of ABC transporters, in particular, P-gp [86,87]. As colon cancer is one of the solid tumors that most frequently show MDR, a MDR subcell line that was characterized by P-gp overexpression was developed from the human colon cancer HT-29 cell line. As expected, Hyp-SDT was able to significantly decrease MDR cancer cell proliferation, while this was unaffected by doxorubicin, which is one of the most frequently used chemotherapeutic drugs and is subjected to transporter-mediated MDR [88].

In conclusion, this work demonstrates Hyp responsiveness to US and its ability to act as a sonosensitizer that can induce, under US exposure, selective and efficient cancer cell killing. US also has the potential to elicit the multifaceted anticancer activity of Hyp by provoking a specific anticancer immune response and hindering acquired MDR. Hypericin-mediated sonodynamic treatment has been investigated in both 2D and 3D colorectal cancer cell cultures, providing useful data on the potential application of this approach in the in vivo setting.

## Figures and Tables

**Figure 1 pharmaceutics-14-01102-f001:**
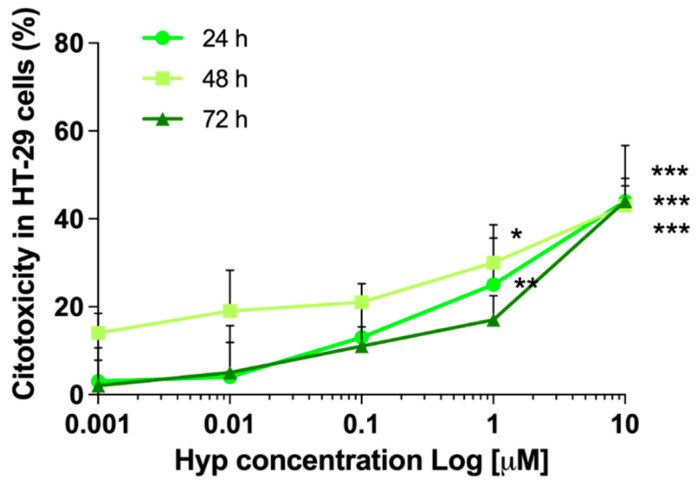
Hypericin cytotoxicity in HT-29 2D cells. Cytotoxicity curves in HT-29 2D cell cultures were obtained by incubating cells with increasing concentrations of Hyp (0.001, 0.01, 0.1, 1, and 10 μM). WST-1 assay was carried out to evaluate Hyp cytotoxicity that was expressed as a percentage compared with untreated cells after 24, 48, and 72 h of incubation. Statistically significant difference versus untreated cells: * *p* ≤ 0.05, ** *p* ≤ 0.01, and *** *p* ≤ 0.001 (*n* = 2).

**Figure 2 pharmaceutics-14-01102-f002:**
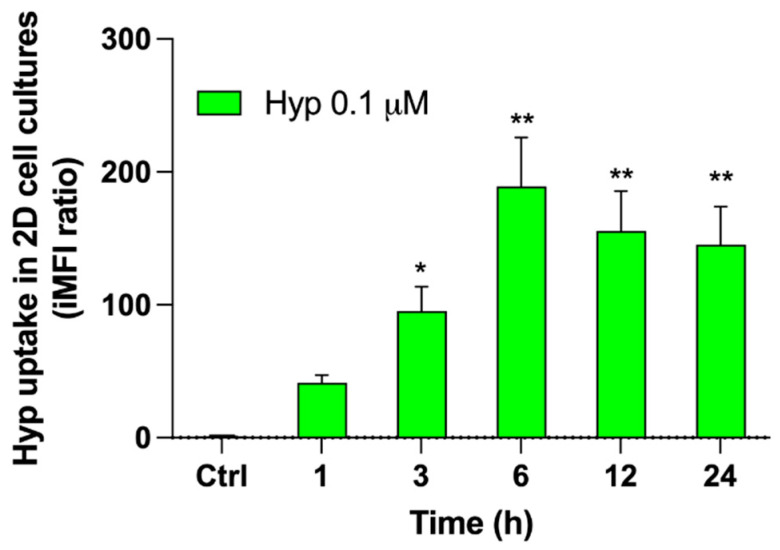
Hypericin uptake in 2D HT-29 cell cultures. HT-29 cells were incubated for 1, 3, 6, 12, and 24 h with 0.1 μM Hyp. Fluorescent signals were detected using a flow cytometer at 488 nm excitation to measure the intracellular amount of Hyp that was expressed as iMFI ratio versus untreated cells (Ctrl). Statistically significant difference versus untreated cells (Ctrl): * *p* ≤ 0.05, ** *p* ≤ 0.01 (*n* = 3).

**Figure 3 pharmaceutics-14-01102-f003:**
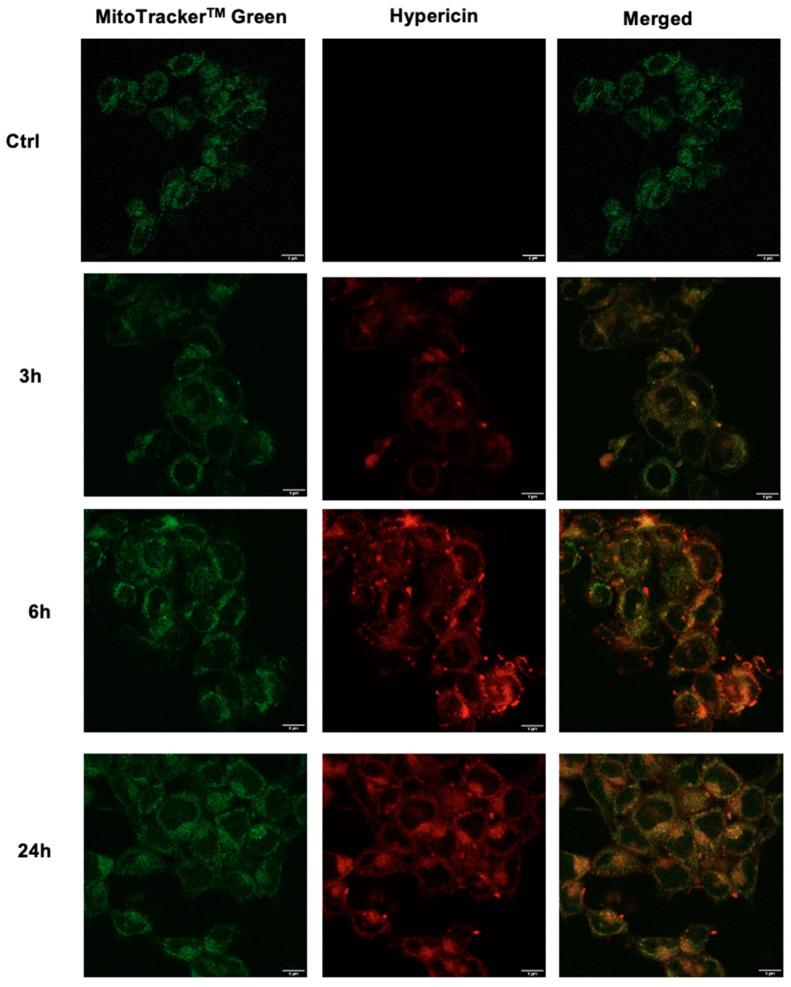
Confocal fluorescence images of HT-29 cells incubated with Hyp 0.1 μM for 3, 6, and 24 h. Pictures in green indicate MitoTrackerTM green used as a mitochondrial counterstain. Pictures in red indicate Hyp. On the right of the panel, merged images indicate the overlay between MitoTracker^TM^ green and Hyp. Magnification: 40×, scale bar: 8 μm.

**Figure 4 pharmaceutics-14-01102-f004:**
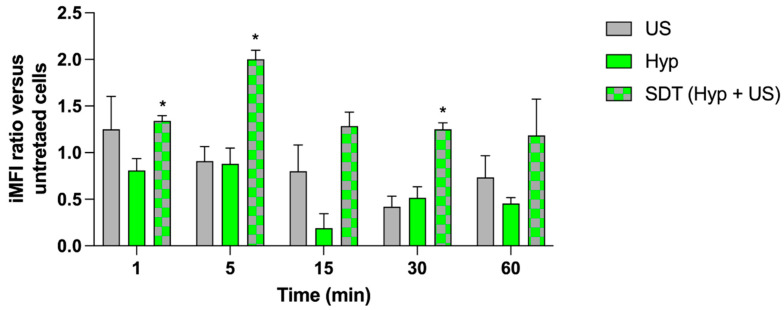
ROS generation under sonodynamic exposure of Hyp. HT-29 cells were incubated for 24 h with 0.1 μM Hyp and then exposed to US (1.5 W/cm^2^ at 1.5 MHz, continuous wave mode for 5 min). Statistically significant difference versus untreated cells: * *p* ≤ 0.05 (*n* = 3).

**Figure 5 pharmaceutics-14-01102-f005:**
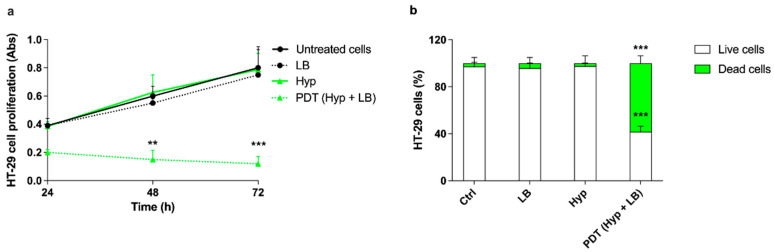
Effects of hypericin-mediated photodynamic treatment on HT-29 cell proliferation. HT-29 cells were incubated for 24 h with 0.1 μM Hyp and then exposed to LB (green LED at 1.5 W/cm^2^ for 20 min). (**a**) Cell proliferation was evaluated by WST-1 assay 24, 48, and 72 h after treatment. Statistically significant difference versus untreated cells: ** *p* ≤ 0.01, *** *p* ≤ 0.001. (**b**) Cell death was evaluated, 48 h after the treatment, using flow cytometry with SYTOX™ green dead cell staining and expressed as cell percentage. Statistically significant difference versus untreated cells (Ctrl): ** *p* ≤ 0.01, *** *p* ≤ 0.001 (*n* = 3).

**Figure 6 pharmaceutics-14-01102-f006:**
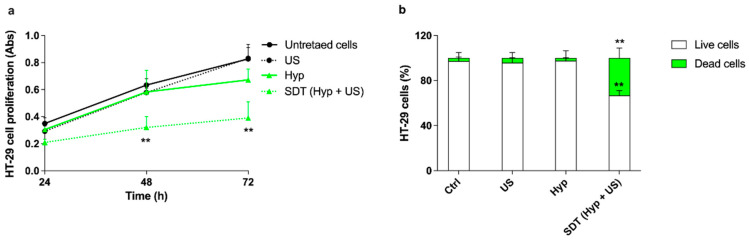
Effects of hypericin-mediated sonodynamic treatment on HT-29 cell proliferation. HT-29 cells were incubated for 24 h with 0.1 μM Hyp and then exposed to US (1.5 W/cm^2^ at 1.5 MHz, continuous wave mode for 5 min). (**a**) Cell proliferation was evaluated by WST-1 assay 24, 48, and 72 h after treatment. Statistically significant difference versus untreated cells: ** *p* ≤ 0.01. (**b**) Cell death was evaluated, 48 h after the treatment, using flow cytometry with SYTOX™ green dead cell staining and expressed as cell percentage. Statistically significant difference versus untreated cells (Ctrl): ** *p* ≤ 0.01 (*n* = 3).

**Figure 7 pharmaceutics-14-01102-f007:**
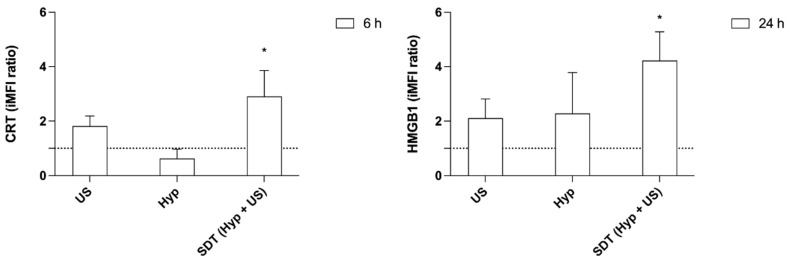
Evaluation of CRT and HMGB1 after hypericin-mediated sonodynamic treatment. HT-29 cells were incubated for 24 h with 0.1 μM Hyp and then exposed to US (1.5 W/cm^2^ at 1.5 MHz, continuous wave mode for 5 min). The presence of CRT and HMGB1 was evaluated 6 and 24 h after treatment, respectively. Results are expressed as iMFI ratio. Statistically significance of treated cells versus untreated cells (dashed line): * *p* ≤ 0.05 (*n* = 3).

**Figure 8 pharmaceutics-14-01102-f008:**
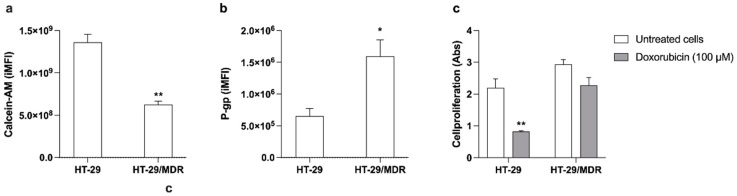
Assessment of the multi-drug-resistant phenotype in HT-29/MDR cells. (**a**) Cellular internalization of a P-gp substrate expressed as the integrated mean fluorescence intensity (iMFI) of calcein-AM, as detected by cytofluorimetric analysis in HT-29 and HT-29/MDR cells. Statistically significant difference versus HT-29 cells: ** *p* ≤ 0.01 (*n* = 3). (**b**) Pg-p expression by cytofluorimetric analysis in HT-29 and HT-29/MDR cells. Statistically significant difference versus HT-29 cells: * *p* ≤ 0.05 (*n* = 3). (**c**) Cell proliferation was evaluated by WST-1 assay after 48 h of doxorubicin (100 μM) incubation. Statistical significance versus untreated cells: ** *p* ≤ 0.01 (*n* = 2).

**Figure 9 pharmaceutics-14-01102-f009:**
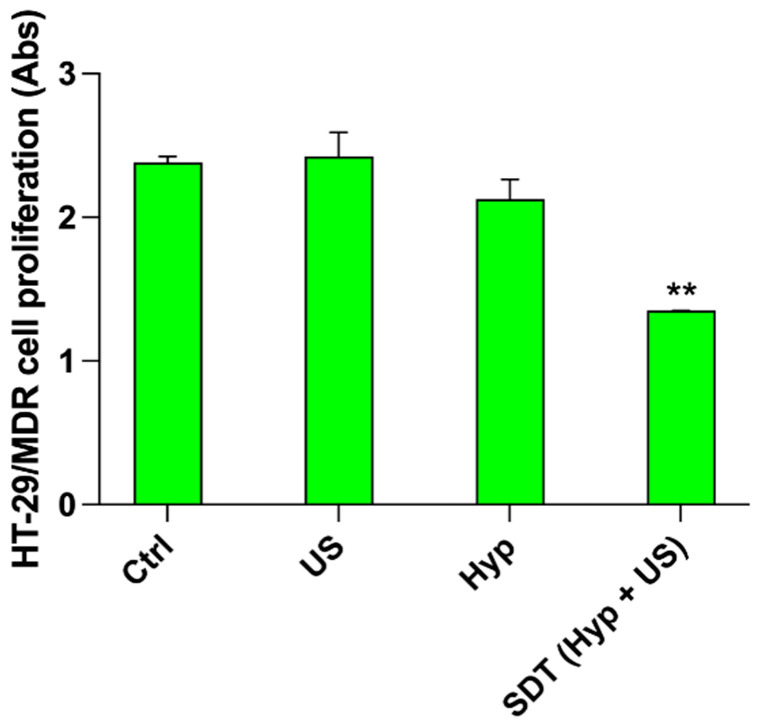
Effects of Hyp-mediated sonodynamic treatment on HT-29/MDR cell proliferation. HT-29 cells were incubated for 24 h with 0.1 μM Hyp and then exposed to US (1.5 W/cm^2^ at 1.5 MHz, continuous wave mode for 5 min). Cell proliferation was then evaluated by WST-1 assay after 48 h. Statistically significant difference versus untreated cells: ** *p* ≤ 0.01 (*n* = 3).

**Figure 10 pharmaceutics-14-01102-f010:**
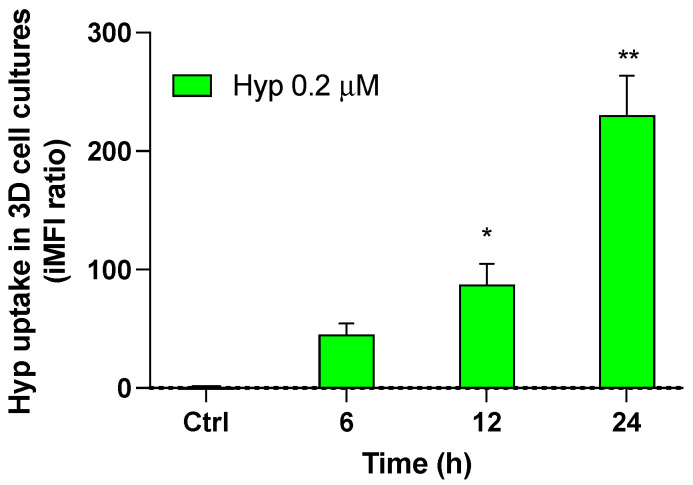
Hypericin uptake in 3D HT-29 cell cultures. HT-29 spheroids were incubated for 24 h with 0.2 μM Hyp. Fluorescent signals were detected using a flow cytometer at 488 nm excitation to measure the intracellular amount of Hyp, which was expressed as iMFI ratio. Statistically significant difference versus untreated cells (Ctrl): * *p* ≤ 0.05, ** *p* ≤ 0.01 (*n* = 3).

**Figure 11 pharmaceutics-14-01102-f011:**
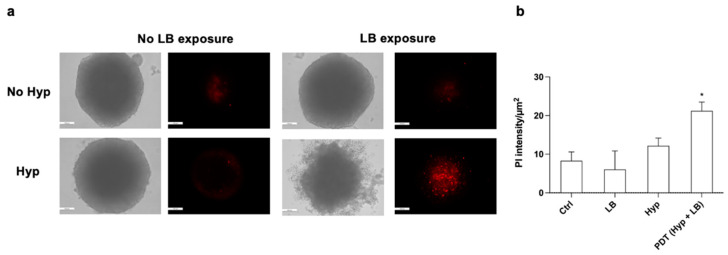
Effects of Hyp-mediated photodynamic treatment on HT-29 spheroids. (**a**) Representative optical (bright field) and fluorescence (PI staining in red) images that were acquired 48 h after the spheroid treatment as follows: untreated spheroid, spheroid incubated with 0.2 µM Hyp, spheroid exposed to LB (green led at 1.5 W/cm^2^ for 20 min), and spheroid treated with PDT (0.2 µM Hyp + LB). Magnification 10×, scale bar: 100 μm. (**b**) Quantification of PI fluorescence intensity in HT-29 spheroids after the different treatments. Statistically significant difference of treated cells versus untreated cells: * *p* ≤ 0.05 (*n* = 3).

**Figure 12 pharmaceutics-14-01102-f012:**
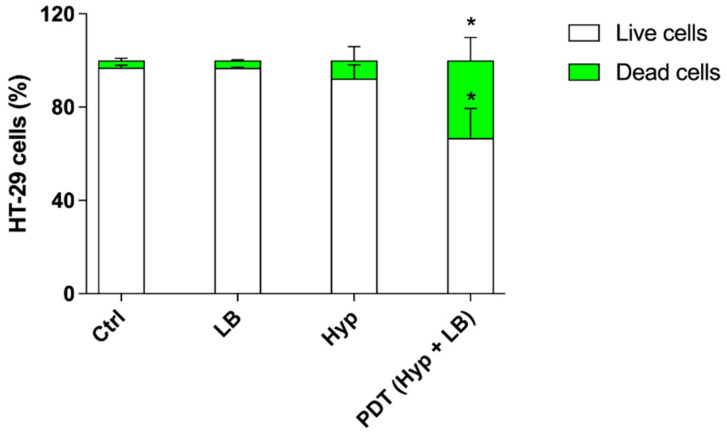
Cell death evaluation after Hyp-mediated photodynamic treatment in HT-29 spheroids. HT-29 spheroids were previously incubated with 0.2 µM Hyp and then exposed to LB (green LED at 1.5 W/cm^2^ for 20 min). Cell death was evaluated by flow cytometry using SYTOX™ green dead cell staining and expressed as cell percentage. Statistically significant difference versus untreated cells (Ctrl): * *p* ≤ 0.05 (*n* = 3).

**Figure 13 pharmaceutics-14-01102-f013:**
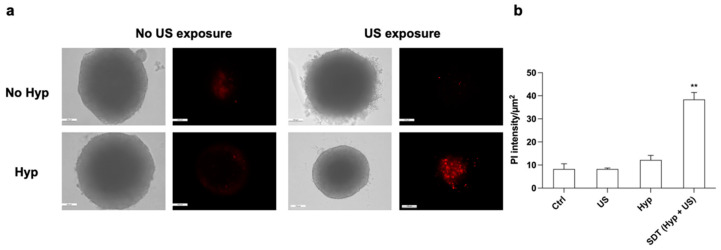
Effects of Hyp-mediated sonodynamic treatment on HT-29 spheroids. (**a**) Representative optical (bright field) and fluorescence (propidium iodide staining in red) images acquired 48 h after the spheroid treatment as follows: untreated spheroid, spheroid incubated with 0.2 µM Hyp, spheroid exposed to US (1.5 W/cm^2^ at 1.5 MHz, continuous wave mode for 5 min), and spheroid treated with SDT (0.2 µM Hyp + US). Magnification: 10×, scale bar: 100 μm. (**b**) Quantification of PI fluorescence intensity in HT-29 spheroids after the different treatments. Statistically significant difference of treated cells versus untreated cells: ** *p* ≤ 0.01 (*n* = 3).

**Figure 14 pharmaceutics-14-01102-f014:**
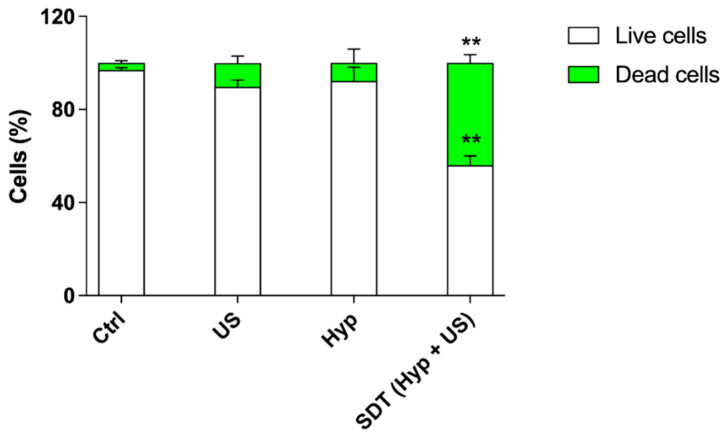
Cell death evaluation after Hyp-mediated sonodynamic treatment in HT-29 spheroids. HT-29 spheroids were previously incubated with 0.2 µM Hyp and then exposed to US (1.5 W/cm^2^ at 1.5 MHz, continuous wave mode for 5 min). Cell death was evaluated by flow cytometry using SYTOX™ green dead cell staining and expressed as cell percentage. Statistically significant difference versus untreated cells (Ctrl): ** *p* ≤ 0.01 (*n* = 3).

## Data Availability

The data presented in this study are available on request from the corresponding author.

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
