# Peer review of "Ultrasound Triggers Hypericin Activation Leading to Multifaceted Anticancer Activity"

_pharmaceutics, 2022, doi:10.3390/pharmaceutics14051102_

Round 1
Reviewer 1 Report
This research is interesting, but some description or discussion is poor. The readers must be confusing without some sentences.
Taken together, major revisions should be made before re-submission. The paper would be re-considered only when all the comments were responded.
- Introduction
The authors should introduce and explain the 3D cancer cell culture to evaluate the therapeutic efficacy in vitro. Without the sections, it is difficult to understand the study. To reduce the authors’ burden, I suggest at least these recent reviews be added for revision. At the current version, the reviewer cannot recommend the publication.
Drug effects Cancers 2020, 12(10), 2754
Nanomedicines Advanced Drug Delivery Reviews 79–80 (2014) 95–106
PDT Nanoscale, 2018, 10, 1570-1581
- Figures 11 and 13
I think the size of spheroids was too large to evaluate the treatment. The cells in the center of spheroids must be dead due to hypoxia.
In addition, it is difficult to see fluorescence clearly.
- Figure 7
Why is the bar green? I think meaningless.
- Figures 9 and 12
To evaluate the cell proliferation inhibition, the authors should investigate the Ki67.
- Discussion
This is an essential part of using the tumor spheroids in this study, right? The information is too poor. The authors should introduce the concept of 3D models and application examples. Are there little papers on PDT by using 3D tumor spheroids? The authors should reveal the points. To reduce the authors’ burden, I suggest the sentences or references to be added for the revision. At the current version, the reviewer cannot recommend the publication.
Sentences
3D cancer models have been recently noted to investigate the cancer ability or therapeutic efficiency, such as growth [], gene alternation [], invasion [], morphology [], or drug resistance []. These systems enabled the promotion of the development of new drug candidates or novel therapeutic effect. However, there are few papers on PDT……….
Proliferation
doi.org/10.1016/j.biomaterials.2018.10.014
Alternation
Cells 2022, 11(2), 305
Invasion
Tissue Eng. Part C Methods 2019, 25, 711–720 https://doi.org/10.1089/ten.tec.2019.0189
Morphology
Scientific Reports volume 9, Article number: 292 (2019)
Drug resistance
doi.org/10.1002/bit.26845
Reviewer 2 Report
In this manuscript, the authors investigate the cytotoxic effect of sonodynamic therapy from a natural sonosensitizer, hypericin, on HT29 colorectal cancer cell lines and spheroids. The manuscript is well-written, and methodology is well-documented and seems to be reproducible. However, some changes needed to be made and a few grammar checked before the manuscript could be accepted for publication.
The authors did an excellent job in listing the statistics in the figure legends and I suggest they also mention how many biological replicates were done on each graph.
Line 105 to 110: the definition of inertial cavitation and non-inertial cavitation is messed up. What you describe is the inertial cavitation is actually non-inertial. Inertial cavitation is the ones involves bubble collapsing.
Line 362-365: the results showed that a markedly increased uptake from 6-24 hours post incubation. From the graph, it seems like 6-hour post incubation has the highest value. Can the authors give more explanation of why 24-hour was chosen instead?
Check line 385. weas --> were? Was?
Line 463-464: the data of increased P-gp was not shown. However, this is very important as it gives the reader a sense of drug resistance by knowing how many folds of P-gp was increased compared to the parental cell line.
The spheroid data are interesting. The authors can also discuss, based on their spheroid data assuming the spheroid sizes were the same when the treatment starts, what is the advantage of SDT over PDT (e.g. penetration? Hypoxia resistance?)? The flow cytometry data suggests more cell kill from SDT compared to PDT. It would be nice to have a cross-section immunofluorescence staining of PI or hypericin in the spheroids to understand the degree of cell death and drug penetration under different spatial locations.
Round 2
Reviewer 1 Report
I recommend the publication